# To tweet or not to tweet, that is the question: A randomized trial of Twitter effects in medical education

**Lauren A. Maggio**[1]*, **Todd C. Leroux**[2], **Anthony R. Artino, Jr.**[1]

**1** Department of Medicine, Uniformed Services University of the Health Sciences, Bethesda, Maryland, United States of America, **2** Defense Health Agency, Department of Defense, Falls Church, Virginia, United States of America

* lauren.maggio@usuhs.edu

## Abstract

### Introduction

Many medical education journals use Twitter to garner attention for their articles. The purpose of this study was to test the effects of tweeting on article page views and downloads.

### Methods

The authors conducted a randomized trial using *Academic Medicine* articles published in 2015. Beginning in February through May 2018, one article per day was randomly assigned to a Twitter (case) or control group. Daily, an individual tweet was generated for each article in the Twitter group that included the title, #MedEd, and a link to the article. The link delivered users to the article's landing page, which included immediate access to the HTML full text and a PDF link. The authors extracted HTML page views and PDF downloads from the publisher. To assess differences in page views and downloads between cases and controls, a time-centered approach was used, with outcomes measured at 1, 7, and 30 days.

### Results

In total, 189 articles (94 cases, 95 controls) were analyzed. After days 1 and 7, there were no statistically significant differences between cases and controls on any metric. On day 30, HTML page views exhibited a 63% increase for cases (M = 14.72, SD = 63.68) when compared to controls (M = 9.01, SD = 14.34; incident rate ratio = 1.63, p = 0.01). There were no differences between cases and controls for PDF downloads on day 30.

### Discussion

Contrary to the authors' hypothesis, only one statistically significant difference in page views between the Twitter and control groups was found. These findings provide preliminary evidence that after 30 days a tweet can have a small positive effect on article page views.

**Data Availability Statement:** All data for this project are available under a CC BY license on Zenodo (http://doi.org/10.5281/zenodo.3461035).

**Funding:** The author(s) received no specific funding for this work.

**Competing interests:** I have read the journal's policy and the authors of the manuscript have the following competing interests: Anthony R. Artino Jr. is the Assistant Editor for AM Last Pages, and Lauren A. Maggio is a member of the Academic Medicine editorial board. We are confirming that our roles do not alter our adherence to PLOS ONE's policies on sharing data and materials.

## Introduction

Recently, the use of social media in medical education has increased [1] with trainees, practitioners, and educators adopting these communication tools to facilitate learning, practice improvement, and knowledge translation [2–5]. In this new environment, many medical education journals also use social media, especially Twitter, to highlight research findings, engage readers, and garner attention for their articles [1, 6, 7]. Some journals have even hired staff or editors who are responsible for tweeting about articles upon their publication and hosting related social media events, like Twitter chats for featured articles [8, 9]. Another approach has been for editors of journals, such as at the *Journal of Graduate Medical Education*, to invite authors to draft tweets for submission with their manuscripts. In principle, it makes good sense that increased social media exposure for an article would translate into increased article dissemination and usage, thus warranting these types of journal-driven social media efforts. However, in medical education, we do not yet know whether this type of social media engagement affects article dissemination, as measured by article-level metrics like page views.

In biomedicine, researchers have explored the question of whether journal-driven Twitter strategies are an effective means of increasing article views [10–12]. To date, findings have been mixed. For example, Fox et. al. conducted a randomized trial of articles published in *Circulation* and found no difference in 30-day page views for articles that were tweeted and posted to Facebook when compared to those that were not [10, 11]. In contrast, a separate group of researchers affiliated with the Cochrane Collaboration tweeted Cochrane Reviews and found a three-fold increase in views for tweeted articles over those that were not [12]. Beyond journal-driven social media efforts, other more comprehensive approaches (e.g., those driven by physicians active on social media and by editorial board members) have been shown to positively affect a journal's web traffic [13–14].

To build on this previous work, and to better understand the impact of journal-driven social media efforts in medical education, we conducted a randomized trial to test the effects of social media engagement—specifically, tweets by a single journal, *Academic Medicine* (AM)—on article page views. We hypothesized that, on average, tweeted articles would receive more page views and more downloads than articles that were not tweeted.

## Method

We worked with AM's professional editorial staff to conduct this randomized, parallel design study in 2018. Eligible AM articles were those published in 2015. We focused on 2015 articles because they are both contemporary and also publicly available on the AM website (i.e., the full texts were not obstructed by a paywall). We excluded articles published after 2015 because AM has a policy to tweet all new articles, and the journal did not want to disenfranchise authors of articles that were not tweeted as part of a study. Because this study did not include human participants, but rather focused on publicly accessible articles as the unit of analysis, we did not seek ethical approval.

For those articles AM published in 2015, we included all articles published as Research Reports, Articles, Innovation Reports, Perspectives, and Literature Reviews. Based on a power analysis to detect an average difference of six views, it was determined that we needed 180 articles (90 articles per study group). Excluded article types included Editorials, Invited Commentaries, New Conversations, and other Special Features (e.g., Letters to the Editor and Last Pages). Articles were randomly assigned to a Twitter (case) or control group using Excel's random number generator by an independent researcher, who is credited for her role in the acknowledgements. This researcher also composed all tweets and supplied them directly to the

AM staff to be loaded into their instance of HootSuite, a social media management system. The authors did not engage in the randomization process or delivery of the tweets.

For each article in the Twitter group, the tweet included the article title, the hashtag #MedEd, and an Ow.ly link to the article (e.g., ow.ly/br9d30nxVOF). Hashtags are a word or a phrase preceded by a hash sign embedded within a tweet. They are used to classify tweets and make them more discoverable, especially by individuals not personally connected to the account that tweeted. When clicked, the Ow.ly link delivered the user directly to the article's landing page, which included immediate access to the full text on the HTML page and also provided a link for users to download a PDF version of the article, if desired. Importantly, the article tweets generated by AM did not reference the age of the article (i.e., the article's publication date was not immediately obvious to Twitter users).

Beginning on 5 February 2018, daily at noon Eastern Time, a tweet was generated for one article per day and was automatically posted via Hootsuite, a social media management system, to AM's Twitter account (@AcadMedJournal). Individual article tweets continued until all articles in the Twitter group were tweeted (10 May 2018). After the trial commenced there were no changes to the study methods.

We extracted HTML page view and PDF download count data directly from Wolters Kluwer's (AM's publisher) Adobe Analytics interface. The data for this study are available at 10.5281/zenodo.3461035. To assess the differences in page view and download counts between cases and controls, we used a time-centered approach with outcomes measured at 1, 7, and 30 days. All page view and download counts were cumulative, and we selected these time points based on related Twitter research conducted by Fox and Adams [10, 12].

For each article at each time point, we assessed two user activity count metrics: (1) HTML full-text page views and (2) PDF downloads. All metrics were collected in compliance with COUNTER [15], the international standard for metrics reporting followed by the majority of scholarly publishers, including Wolters Kluwer. An HTML full-text page view was logged if a user visited an article's main page on the AM website. This was the default experience for a user clicking the link in the tweet; it allowed users to directly view a full-text version of the article on the web page. A PDF download was logged if a user clicked either the PDF icon or the link "Article as PDF" on the article's AM web page. We also identified for each article, using Web of Science, the number of times that it had been cited since its publication.

Because of the over-dispersion of the dependent variables (page view and download counts), where the standard deviation exceeds the mean, we fit negative binomial regression models to explore the association between the page view and download metrics and the classification of the manuscript (case or control) at various points in time (1, 7, and 30 days). Model coefficients were converted to incident rate ratios (by taking the exponent of the coefficient), to facilitate a more interpretable result. In this study, incident rate ratios can be interpreted as the expected difference between cases and controls, either positive or negative, specified as a rate. Statistical analysis was performed using R (Version 3.3.3 "Another Canoe") [16] along with packages ggplot2 [17] and MASS [18].

## Results

In total, AM published 417 articles in 2015. Of these 189 articles (94 cases, 95 controls) were analyzed. Analyzed articles represented the following publication types: Articles (cases n = 13, 14%; controls n = 12, 13%), Innovation Reports (cases n = 9, 10%; controls n = 15, 16%), Perspectives (cases n = 22, 23%; controls n = 26, 27%), Research Reports (cases n = 47, 50%; controls n = 38, 40%), Literature Reviews (cases n = 3, 3%; controls n = 4, 4%). Overall, articles in

**Table 1. Descriptive Statics Stratified by Page-View Metrics, Time Period, and Condition for a 2018 Study of the Effects of Journal Tweeting on article page views.**

| | | HTML Full-Text Views | | | | | PDF Full-Text Views | | | | |
|---|---|---|---|---|---|---|---|---|---|---|---|
| | | *n* | *Mean* | *SD* | *Median* | *IQR* | *n* | *Mean* | *SD* | *Median* | *IQR* |
| **Day 1** | | | | | | | | | | | |
| | *Control* | 95 | 0.56 | 1.18 | 0 | 1 | 95 | 0.38 | 0.90 | 0 | 0 |
| | *Case (Tweet)* | 94 | 0.91 | 3.25 | 0 | 1 | 94 | 0.28 | 0.71 | 0 | 1 |
| **Day 7** | | | | | | | | | | | |
| | *Control* | 95 | 2.03 | 3.79 | 1 | 3 | 95 | 0.85 | 1.35 | 0 | 1 |
| | *Case (Tweet)* | 94 | 2.85 | 8.02 | 1 | 3 | 94 | 1.09 | 1.84 | 0 | 1 |
| **Day 30** | | | | | | | | | | | |
| | *Control* | 95 | 9.01 | 14.34 | 4 | 8.5 | 95 | 4.22 | 5.88 | 2 | 3 |
| | *Case (Tweet)* | 94 | 14.72 | 63.68 | 5 | 7.75 | 94 | 5.12 | 7.73 | 2 | 4 |

Note: SD = standard deviation; IQR = interquartile range

our sample were cited on average 19.82 times (cases = 21.39; controls = 18.13) ranging from zero to 127 citations.

All tweets originated from the AM Twitter account, which featured, on average, 7,951 followers (range during the time of the study: 7,612–8,282). Table 1 provides summary statistics stratified by page view and download metrics at various points in time (1, 7, and 30 days) and category of interest (case or control). In general, the mean and median for most user activity metrics was quite low (counts were often less than 5).

Table 2 contains summary information from the negative binomial regression models. As shown in the table, after days 1 and 7 of tweets, there were no statistically significant differences between cases and controls on any of the page view or download metrics. On day 30, however, tweeted articles attained 63% more HTML full-text page views (M = 14.72, SD = 63.68) than controls (M = 9.01, SD = 14.34; incident rate ratio = 1.63, p = 0.01). There were no statistically significant differences between cases and controls for PDF downloads on day 30.

**Table 2. Negative Binomial Regression Model Results Stratified by Page-View Metrics, Time Period, and Condition for a 2018 Study of the Effects of Journal Tweeting on article page views.**

| | | HTML Full-Text Views | | | PDF Full-Text Views | | |
|---|---|---|---|---|---|---|---|
| | | *IRR* | *P-value* | *IRR 95% CI* | *IRR* | *P-value* | *IRR 95% CI* |
| **Day 1** | | | | | | | |
| | *Control* | *Ref* | | - | - | - | - |
| | *Case (Tweet)* | 1.60 | 0.14 | 0.84–3.06 | 0.73 | 0.39 | 0.36–1.48 |
| **Day 7** | | | | | | | |
| | *Control* | *Ref* | | - | - | - | - |
| | *Case (Tweet)* | *1.40* | 0.13 | 0.90–2.18 | 1.28 | 0.31 | 0.79–2.09 |
| **Day 30** | | | | | | | |
| | *Control* | *Ref* | | - | - | - | - |
| | *Case (Tweet)* | 1.63 | 0.01 | 1.10–2.41 | 1.21 | 0.30 | 0.83–1.76 |

Note: IRR = Incident Rate Ratio; IRR CI = Incident Rate Ratio 95% Confidence Interval; Ref. = reference group

## Discussion

Although Twitter has been celebrated as a channel for connecting individuals with research [19], contrary to our hypothesis, we found only one statistically significant difference between the Twitter and control groups in the present study. However, in light of the rigorous experimental design employed here, we believe this is likely a causal relationship, and we encourage other researchers to attempt to replicate these findings in future studies. What is more, of the comparisons that were not statistically significantly different, all but one of the point estimates trended in the expected direction, with Twitter group averages slightly above control group averages. Considering the fairly simple nature of the social media strategy employed here—that is, a single journal tweet featuring only the article's title and #MedEd to promote a given article—these results are promising for journals hoping to use Twitter to improve article dissemination. Use of more comprehensive social media strategies, and their effects on article dissemination, should be tested in a similarly rigorous manner using a broader sample of medical education journals.

This investigation adds to the ongoing study of the efficacy of journal-driven social media strategies, the findings of which have been mixed in previous work. Our results align with those of Adams [16], suggesting that journal-led approaches can have some limited impact on page views. Journal editors and editorial board members should consider these findings when designing their own social media strategies and making decisions on the allocation of resources for such purposes. Furthermore, for medical education researchers exploring their publication options, there may be value in first determining whether or not a given journal maintains a social media presence and the impact that such presence might have on their article's dissemination. A study of medical education journals recently reported that out of 13 core medical education journals, only five had Twitter accounts [1].

The findings reported here should be considered in relation to other complementary social media approaches. For example, Hawkins et. al. [14] recently demonstrated that enlisting editorial board members and their trainees to tweet articles can increase clicks to articles. In addition, it is worth noting that journals are not alone in driving social media attention. For instance, a recent study examined the effects of a physician-led program that enlisted teams of physicians active on social media to tweet articles [13], and another study considered the impact of article infographics and podcasts on article views [20]. To consider and measure the impact of these approaches, researchers in urology have proposed the use of the Twitter Impact Factor (TIF), which, similar to the journal impact factor, is based on citation metrics [21]. These various approaches can inform journal editors as they deliberate on how best to bundle social media outreach to positively affect article dissemination and use. That said, more rigorous research is needed to explore the effects of these approaches, including the practice of enlisting editorial board members and authors, as well as adding other Twitter handles (e.g., of the article's authors and/or other users with large followership) as a way of potentially amplifying the impact of article tweets.

Researchers in public health have also investigated the impact of Twitter hashtags and have found that hashtags can significantly amplify an organization's message, especially if multiple hashtags are used and if they incite action [22]. As described in the Method, Hashtags are a word or a phrase preceded by a hash sign embedded within a tweet (e.g., #assessment, #MedEdChat, #OpenScience). Hashtags are used to classify tweets and make them more discoverable, especially by individuals who do not follow the account that posted the original tweet. Our intervention incorporated only one hashtag, #MedEd, in all of the tweets. Additionally, related to the content of the disseminated tweets, this study employed a rather simplistic

approach, which included only the title of the article. If the tweets had included more targeted hashtags or more descriptive text highlighting or summarizing the main findings of the article, it is possible that such an approach could have increased user engagement. Future research should examine the role of hashtags and descriptive tweets in medical education to better understand how Twitter can be used to facilitate information sharing.

Our study has several important limitations. To begin, we focused on a single journal, AM. Had we investigated a different medical education journal, or group of journals, we may have observed different results. Next, our study relied on data collected from AM's publisher, Wolters Kluwer. Some readers might consider this to be a conflict of interest. However, two points are worth mentioning. First, Wolters Kluwer is a signatory of the COUNTER Code of Practice, which pledges consistent, credible, and comparable production of article usage metrics [15]. Second, Wolters Kluwer and the AM staff played no role in the analysis of these data.

A third important limitation of the present study is that we focused on articles published in 2015, three years prior to initiation of our study. We suspect that the age of these articles may have negatively affected user engagement of articles in both study arms, with older articles potentially being less interesting and/or relevant to readers than newer articles (and thus less likely to be viewed). That said, it is important to note that the tweets created in this study did not reference the age of the article, and so Twitter users would have no way of knowing these were 2015 articles until they clicked on the article link (unless, of course, they were already familiar with the article title). Nonetheless, more work is needed to investigate the impact of tweeting on contemporary articles.

In this study, we did not examine the impact of an article's publication type; for example, does a literature review garner more views when tweeted than an innovation report? Although beyond the scope of the current study, we encourage researchers to consider the effects of publication types in future studies. Additionally, due to the relatively low volume of Twitter activity with the manuscripts included in the case and control samples, the fixed sample size only detected differences in Twitter activity amounting to moderate differences. These differences, however, are practically meaningful to researchers and journal staff actively engaged in marketing articles via social media.

Finally, while we assumed that increased clicks to an article's journal web page translate into more engagement with article content (i.e., users actually reading the article), we did not specifically test this assumption and we found no statistically significant differences at all time points for PDF downloads. This finding may suggest that deeper engagement with the tweeted articles–for example, taking the extra step of downloading the PDF–was not stimulated by the tweet approach tested here.

## Conclusion

Results from this randomized trial revealed that after 30 days a tweet can have a small positive effect on the number of HTML page views an article receives, increasing those views by 63%. While small, this increase may still be meaningful, especially if one considers the simple intervention studied here (i.e., a single, text-based tweet from a single journal). In light of these results, journal editors may want to consider using Twitter as a means of improving dissemination, but they should also contemplate the best ways to combine simple approaches with other, more robust social media strategies. In addition, researchers should work to better understand if and how Twitter and other social medical strategies can be used to improve article dissemination.

## Acknowledgments

Disclaimer: The views expressed in this article are those of the authors and do not necessarily reflect the official policy or position of the Uniformed Services University of the Health Sciences, the U.S. Navy, the Department of Defense, or the U.S. Government.

The authors wish to thank the *Academic Medicine* editorial staff members for their assistance in designing the study and supplying access to the data. The authors also thank Dr. Holly Meyer for her assistance in the conceptualization of this study and her participation in the randomization process and delivery of the tweets to the journal.

## Author Contributions

**Conceptualization:** Lauren A. Maggio, Todd C. Leroux, Anthony R. Artino, Jr.

**Data curation:** Lauren A. Maggio, Todd C. Leroux, Anthony R. Artino, Jr.

**Formal analysis:** Lauren A. Maggio, Todd C. Leroux, Anthony R. Artino, Jr.

**Investigation:** Lauren A. Maggio, Todd C. Leroux, Anthony R. Artino, Jr.

**Methodology:** Lauren A. Maggio, Todd C. Leroux, Anthony R. Artino, Jr.

**Project administration:** Lauren A. Maggio.

**Writing – original draft:** Lauren A. Maggio, Todd C. Leroux, Anthony R. Artino, Jr.

**Writing – review & editing:** Lauren A. Maggio, Todd C. Leroux, Anthony R. Artino, Jr.

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
