## [Decision Letter · Decision Letter 0]

16 Sep 2019

PONE-D-19-19000

To Tweet or Not to Tweet, That is the Question: A Randomized Trial of Twitter Effects in Medical Education

PLOS ONE

Dear Dr. Maggio,

Thank you for submitting your manuscript to PLOS ONE. After careful consideration, we feel that it has merit but does not fully meet PLOS ONE’s publication criteria as it currently stands. Therefore, we invite you to submit a revised version of the manuscript that addresses the points raised during the review process.

We would appreciate receiving your revised manuscript by Oct 31 2019 11:59PM. To enhance the reproducibility of your results, we recommend that if applicable you deposit your laboratory protocols in protocols.io, where a protocol can be assigned its own identifier (DOI) such that it can be cited independently in the future. For instructions see: http://journals.plos.org/plosone/s/submission-guidelines#loc-laboratory-protocols

We look forward to receiving your revised manuscript.

Kind regards,

Luigi Lavorgna

Academic Editor

PLOS ONE

Journal Requirements:

Reviewers' comments:

Reviewer's Responses to Questions

**Comments to the Author**

1. Is the manuscript technically sound, and do the data support the conclusions?

Reviewer #1: Partly

Reviewer #2: Yes

2. Has the statistical analysis been performed appropriately and rigorously? 

Reviewer #1: No

Reviewer #2: Yes

3. Have the authors made all data underlying the findings in their manuscript fully available?

Reviewer #1: Yes

Reviewer #2: No

4. Is the manuscript presented in an intelligible fashion and written in standard English?

Reviewer #1: Yes

Reviewer #2: Yes

5. Review Comments to the Author

Reviewer #1: Maggio and colleagues studied the impact of Tweets on medical education and found a minimal increase in page views for tweeted articles. Overall, the paper is clear and well written. The effort of doing a randomized trial on this topic is commendable. However, I have some methodological concerns I would like the authors comment on.

Authors state that: “We excluded articles published after 2015 because AM has a policy to tweet all new articles, and the journal did not want to disenfranchise authors of articles that were not tweeted as part of a study”. The same applies to placebo in clinical trials but does not prevent from doing placebo-controlled trials. It would be good if authors comment on that. E.g., three-year old articles might attract less attention independently from Tweets.

Authors based the sample size calculation on a difference of 6 views that corresponds to 40-65% effect size based on 30-day HTML Full-Text Views and >100% effect size based on 30-day PDF Full-Text Views. As such, the study is underpowered and the lack of statistical significance is not surprising, especially for Day 1, Day 7 and PDF Full-Text Views. Authors should consider expanding the sample or, at least, carefully commenting on this in the Discussion (and toning it seriously down). Indeed, the only statistically significant measure is the one with reasonable effect size (40% for 30-day HTML Full-Text Views).

It would be good to known the number of citations of included articles in the two groups.

Authors might find interesting commenting on Eysenbach G, J Med Internet Res 2011, and Cardona-Grau D et al, Eur Urol Focus 2016.

Reviewer #2: The paper is original covering newaspects regarding medical education and the effect of social media on it.

Although the comparison between the two groups show statistically difference the authors should better define the type of article that have been downloaded (i.e. original papers , case reports short communications, review articles) in relation to the different times of observation. Moreover, the authors should clarify why they conducted the study in the 2018 but the article tweetted referred to 2015. One limitation of the study which should be discussed in the paper is the selection of the tweet indicating only the title of the article. A more appealing choice probably could increse the attention of followers.

6. PLOS authors have the option to publish the peer review history of their article (what does this mean?). If published, this will include your full peer review and any attached files.

Reviewer #1: No

Reviewer #2: No

---

## [Author Response · Author response to Decision Letter 0]

30 Sep 2019

Please see the included Word document that responds directly to the reviewers' comments.

---

## [Editor Report · Decision Letter 1]

3 Oct 2019

To Tweet or Not to Tweet, That is the Question: A Randomized Trial of Twitter Effects in Medical Education

PONE-D-19-19000R1

Dear Dr. Maggio,

We are pleased to inform you that your manuscript has been judged scientifically suitable for publication and will be formally accepted for publication once it complies with all outstanding technical requirements.

With kind regards,

Luigi Lavorgna

Academic Editor

PLOS ONE
---

## [Editor Report · Acceptance letter]

8 Oct 2019

PONE-D-19-19000R1 

To Tweet or Not to Tweet, That is the Question: A Randomized Trial of Twitter Effects in Medical Education 

Dear Dr. Maggio:

I am pleased to inform you that your manuscript has been deemed suitable for publication in PLOS ONE. Congratulations! Your manuscript is now with our production department. 

With kind regards,

on behalf of

Dr. Luigi Lavorgna 

Academic Editor

PLOS ONE